# Focus on DNA Glycosylases—A Set of Tightly Regulated Enzymes with a High Potential as Anticancer Drug Targets

**DOI:** 10.3390/ijms21239226

**Published:** 2020-12-03

**Authors:** Fabienne Hans, Muge Senarisoy, Chandini Bhaskar Naidu, Joanna Timmins

**Affiliations:** Institut de Biologie Structurale (IBS), University Grenoble Alpes, CEA, CNRS, F-38000 Grenoble, France; fabienne.hans@ibs.fr (F.H.); muge.senarisoy@i2bc.paris-saclay.fr (M.S.); chandinipreethi21@gmail.com (C.B.N.)

**Keywords:** cancer, base excision repair, DNA glycosylases, drug resistance, inhibitors

## Abstract

Cancer is the second leading cause of death with tens of millions of people diagnosed with cancer every year around the world. Most radio- and chemotherapies aim to eliminate cancer cells, notably by causing severe damage to the DNA. However, efficient repair of such damage represents a common mechanism of resistance to initially effective cytotoxic agents. Thus, development of new generation anticancer drugs that target DNA repair pathways, and more particularly the base excision repair (BER) pathway that is responsible for removal of damaged bases, is of growing interest. The BER pathway is initiated by a set of enzymes known as DNA glycosylases. Unlike several downstream BER enzymes, DNA glycosylases have so far received little attention and the development of specific inhibitors of these enzymes has been lagging. Yet, dysregulation of DNA glycosylases is also known to play a central role in numerous cancers and at different stages of the disease, and thus inhibiting DNA glycosylases is now considered a valid strategy to eliminate cancer cells. This review provides a detailed overview of the activities of DNA glycosylases in normal and cancer cells, their modes of regulation, and their potential as anticancer drug targets.

## 1. Introduction

Cancer is expected to rank as the leading cause of death and the major barrier to increased life expectancy in the 21st century [1]. Carcinogenesis is considered to be a multistep process leading to the formation of complex, dynamic, and heterogeneous tumoral tissues containing distinct cell types [2]. One of the characteristics of tumor cells is to divide in an unregulated way, enabling tumors to grow rapidly and invade normal tissues [2]. Cancer cells have also been reported to produce high levels of reactive oxygen species (ROS) [3] and to present dysregulated and/or impaired DNA repair systems, both of which lead to an accumulation of high levels of DNA damage in both genomic and mitochondrial DNA [4,5,6,7,8,9]. To survive, tumors take advantage of their extraordinary genomic plasticity, which can lead to abnormalities in chromosome copy numbers (aneuploidy or polyploidy) and chromosome rearrangements (e.g., translocation or chromothripsis), but also to altered cellular and subcellular compartments, as in multinucleated giant polyploid cells that are known to contribute to tumorigenesis, metastasis, and drug resistance [10,11,12,13]. 

With the exception of surgery, all major anticancer therapeutic strategies, including radio-, chemo-, and immuno-therapies, used alone or in combination, aim to specifically kill cancer cells within affected tissues [14,15,16]. Cytotoxic chemotherapy, developed 60 years ago, still represents a widely used treatment and in many cases involves the administration of powerful genotoxic agents acting either directly (e.g., platinum-based drugs) or indirectly (e.g., topoisomerase inhibitors) on DNA in cells that rapidly proliferate [17,18,19]. Depending on the level of DNA damage, the cell type, and the state of the cell, genotoxic therapies can induce several cellular responses [20]. Low levels of DNA damage activate the DNA damage response (DDR), leading to cell cycle arrest and stimulation of the DNA repair machinery to maintain genome stability. If repair is incomplete, due to impaired DNA repair systems for instance, DNA damage may generate mutations or chromosome aberrations, which can potentially lead to tumorigenesis. High levels of DNA damage, generated for example by conventional cancer therapies, induce a variety of cell death programs, including apoptosis, necrosis, or senescence, to eliminate the most injured cells. In some cases, even after induction of apoptosis, tumor cells have been shown to recover at later stages, a phenomenon known as anastasis [21]. These diverse cellular responses make successful cancer treatment difficult. 

In addition, DNA repair genes have been shown to be overexpressed in many cancers to reinforce the repair capacity of the tumor [22]. This provides the tumors with an evolutionary advantage compared to normal tissues and appears to be a factor favoring metastasis [23]. This explains why efficient repair of lesions generated by genotoxic therapeutic agents before they become toxic is one of the multiple mechanisms used by tumor cells to develop resistance to initially effective cytotoxic therapy [24,25,26]. As a result, development of drugs that target DNA repair pathways has attracted much attention over the recent years in the cancer drug discovery field and new innovating anticancer strategies targeting DNA repair proteins have emerged [14,25,27,28,29,30]. Success with the inhibitors of poly(ADPribose) polymerase 1 (PARP1) in particular have paved the way for the acceptance of the inhibition of these proteins as a novel concept for cancer therapy [31,32,33]. A majority of these new therapies aims to inhibit the catalytic activity of DNA repair enzymes and/or trap enzyme-DNA complexes. They may be used as adjuvants, either alone or in combination, to enhance the effectiveness of conventional drugs, so as to minimize resistance and negative side effects [30]. Alternatively, new promising anticancer agents have emerged that target DNA repair components as monotherapy, taking advantage of the high frequency of DNA repair defects observed in human cancers [28]. Because of the partial redundancy of DNA repair pathways, inhibition of the remaining functional pathways in DNA repair defective cancer cells should in some cases have a greater impact on the tumor than on normal tissues, thereby improving efficacy and reducing toxicity of the drugs. The major strategy to achieve such selective tumor cell killing has been the principle of synthetic lethality: Defects in either of two genes or proteins have no effect on survival, but combining the two defects results in cell death [25,34,35,36,37]. The principle of synthetic lethality is now successfully applied in clinic for treating *BRCA*-associated ovarian cancers with PARP inhibitors [38].

Mammalian cells have evolved five major elaborate DNA damage repair pathways to detect and repair the wide diversity of DNA lesions, namely base excision repair (BER), nucleotide excision repair (NER), homologous recombination (HR), non-homologous end-joining (NHEJ), and mismatch repair. Each pathway is responsible for the repair of a different type of DNA damage and together they maintain genomic integrity and stability [29,39,40]. There are several ways in which base lesions are generated in cancer cells. Base lesions can be intrinsically acquired due to altered metabolic pathways that produce an excess of ROS, leading to oxidation of DNA bases. Base lesions can also be caused directly by chemotherapeutic agents reacting with DNA as in the case of alkylating agents, and can be generated indirectly by oxidative stress provoked by chemotherapeutic drugs such as cisplatin. By repairing all of these DNA lesions, the BER pathway reduces the cytotoxic effects of anticancer drugs, thereby contributing to the survival of cancer cells [41,42]. As a result, BER enzymes are increasingly considered as valid targets for cancer treatment [14,43,44,45]. This review presents the recent developments in this area with a special focus on DNA glycosylases that initiate the BER pathway.

## 2. DNA Glycosylases and BER

BER is an essential DNA repair pathway that contributes to the stability of the genome by eradicating the vast number of small, non-helical-distorting base lesions from the genome, resulting from oxidative, alkylating, and deamination events, induced by environmental factors, endogenous factors such as ROS, or anticancer agents such chemotherapeutic drugs or radiotherapy [42,46]. Among the damaged bases repaired by the BER pathway (Table 1; recently reviewed in [41]), one can cite the highly produced and mutagenic oxidized base 8-oxo-guanine (8-oxo-G), the lethal thymine glycols (Tg) caused by oxidation of thymine bases, which induce replication and transcription blockade, uracil misincorporation in DNA, abasic sites, and single-strand breaks (SSB) [43,47].

The BER pathway processes damaged bases in a series of successive reactions to eliminate all the intermediate products that otherwise block replication (Figure 1). The BER pathway is initiated by a set of enzymes, known as DNA glycosylases, responsible for identifying and eliminating the damaged bases, thereby generating an apurinic/apyrimidinic (AP) site [48,49,50]. The subsequent BER enzymes then repair the AP-containing DNA damage [42]. An AP endonuclease (APE1) or an AP lyase cleaves the DNA backbone and produces either a single-stranded DNA (ssDNA) nick 5′ to the AP site in the case of AP endonucleases, or 3′ to the AP site for AP lyases (Figure 1). The DNA ends are then processed by one of two sub-pathways, short- or long-patch repair, during which the DNA polymerase fills the gap with the correct nucleotides and the repair mechanism is completed after sealing of the nick by a DNA ligase [51]. These end processing steps take place by different mechanisms depending on the type of DNA glycosylase, the physiological state of the cell, and the availability of BER factors [42]. In short-patch BER, which is used in proliferating and non-proliferating cells, an AP endonuclease removes the 3′-deoxyribose phosphate (dRP) after strand cleavage and a polynucleotide kinase/phosphatase (PNKP) removes the 3′-phosphate, and the single nucleotide gap is then filled and ligated by DNA polymerase β and DNA ligase I or III [42,46]. In addition, PARP1 and XRCC1 participate in some types of short-patch repair. Long-patch BER, which takes place mainly in proliferating cells, makes use of several additional replication proteins, to fill 2–10 nucleotide gaps. These include DNA polymerase δ/ε, PCNA, the flap endonuclease FEN1, and DNA ligase I [42,46]. 

DNA glycosylases bind to damaged bases and then induce the aberrant base to flip out of the double helix and enter the binding site of the enzyme [49,52,53]. The DNA glycosylase then catalyzes the cleavage of the N-glycosidic bond between the substrate base and the 2′-deoxyribose to efficiently remove the damaged base [50,51]. DNA glycosylases can be subdivided into either monofunctional or bifunctional enzymes depending on their catalytic activities (Figure 1 and Table 1). Monofunctional DNA glycosylases like uracil-N glycosylase (UNG) exhibit only DNA glycosylase activity and produce an AP-site, which is further processed by an AP endonuclease, APE1, in humans. In contrast, bifunctional glycosylases, like endonuclease III-like 1 (NTH1) and endonuclease VIII-like 1-3 (NEIL1-3) glycosylases, exhibit both DNA glycosylase and AP-lyase activities. After release of the damaged base, bifunctional DNA glycosylases with a β-lyase activity form a transient Schiff base intermediate between the DNA and an active site lysine residue, after which the sugar-phosphate backbone is nicked via a β-elimination reaction to create a 3′ α,β-unsaturated aldehyde and a 5′ phosphate group [54,55] that are further processed by APE1 and PNKP, respectively. An amino-terminal proline residue provides NEIL1 and NEIL2 with an additional β-δ-elimination activity and makes APE1, but not PNKP, dispensable in these cases [56,57].

DNA glycosylases have been classified into four superfamilies (SF) that reflect their functional activities and their structural features (Figure 1 and Table 1) [42,58,59]. SF1 comprises uracil DNA glycosylases (UDG), such as UNG, single-strand-specific monofunctional uracil DNA glycosylase 1 (SMUG), and thymine DNA glycosylase (TDG). These glycosylases possess a characteristic α/β fold and target the removal of uracil (U) formed by the deamination of cytosine caused by oxidative stress. UNG and SMUG1 have similar substrate specificity as they both remove mis-incorporated U in DNA. UNG1 and UNG2 are two splice variants of *UNG*, which localize respectively to mitochondria and the nucleus [60]. Beyond their role in the error-free repair of U in DNA, UNG2, and also as shown more recently, UNG1, play a key, mutagenic role in somatic hypermutation and class switch recombination during B cell receptor/antibody maturation, two processes that involve deamination of cytidines to uridines in ssDNA [61,62,63]. TDG is capable of removing oxidized or deaminated pyrimidine bases. It has also been reported to participate in the epigenetic demethylation signaling pathway as part of the multistep pathway that removes the methyl group of cytosine within CpG sites [64,65]. The second superfamily (SF2) comprises DNA glycosylases with a characteristic helix-hairpin-helix (HhH) motif, which include glycosylases such as NTH1, 8-oxo-G DNA glycosylase 1 (OGG1), MutY homolog DNA glycosylase (MUTYH), and methyl-CpG-binding protein 4 (MBD4). This family of glycosylases targets various lesions caused by oxidative stress including 8-oxo-G, Tg, 4,6-diamino-5-formamidopyrimidine (FapyA), and 2,6-diamino-4-hydroxy-5-formamido-pyrimidine (FapyG) (Table 1). SF3 comprises only one member, which is the 3-methyladenine-DNA glycosylase also called 3-methyl-purine glycosylase (MPG). Unlike the other superfamilies, MPG is not characterized by a specific fold and targets the damage caused by alkylation instead of oxidative stress. In addition to excising 3-methyladenine (3MeA), MPG also excises DNA bases with methyl or certain other alkyl groups at the N7 and N3 positions of both adenine and guanine [66]. Finally, the fourth superfamily (SF4) comprises NEIL glycosylases, including NEIL1, NEIL2, and NEIL3. The characteristic fold of this superfamily is a helix-two-turn-helix (H2TH) motif. This superfamily is similar to HhH SF2 in function as it targets the oxidative base damages such as FapyG and FapyA [67]. Within each of these four superfamilies, each enzyme displays a distinct, but partially overlapping substrate specificity (Table 1). This partially explains why most DNA glycosylase knockout mice are viable and do not present clear phenotypes, except for *TDG* null mice that show an embryonic lethal phenotype, most likely as a result of its impaired epigenetic function [59,68]. Some DNA glycosylases like UNG, SMUG1, MPG, and NEIL1 can recognize damaged bases both in double-stranded (dsDNA) and in ssDNA, whereas DNA glycosylases like TDG, MBD4, OGG1, and MUTYH only recognize base lesions in dsDNA. 

## 3. Regulation of DNA Glycosylases

In addition to having to repair a myriad of DNA lesions and to function in a coordinated manner with the other actors of the BER pathway, DNA glycosylases also have to operate in all types of differentiated tissues, as well as in embryonic tissues, in diverse environmental conditions and during the whole lifetime of organisms. Their abundance and catalytic activities must thus be tightly regulated, both temporally and spatially, and highly coordinated with the other BER enzymes and even with other DNA repair pathways. This is achieved by a panoply of known cellular processes: Genetic polymorphism, alternative splicing, regulated gene expression, post-translational modifications, protein–protein interactions, and as recently revealed in different contexts, cross-talk between different DNA repair pathways (Figure 1). Several of these processes have been shown to be altered in cancer cells, thereby leading to dysregulated DNA glycosylase activity that in turn contributes to enhanced cell survival.

A first level of regulation comes from the numerous tumor-associated variants of DNA glycosylases that have been described (reviewed in [119]), which strongly argue for a genetic linkage between DNA glycosylases and cancer [120]. For instance, the S326C polymorphic variant of OGG1, which is widespread in Caucasian and Asian populations, has been reported to be associated with different types of cancers [121,122,123,124]. Interestingly, this variant has been shown to have reduced DNA glycosylase activity and is more sensitive to oxidation [41,125,126]. A catalytically inactive variant of MUTYH has also been identified in cell extracts from colorectal cancer cells [103]. Several NTH1 and MUTYH variants inherited in an autosomal recessive manner are associated with polyposis development, which can further degenerate into cancerous forms [127,128]. When expressed in MCF10A, a non-transformed human mammary cell line, the catalytically inactive germline D239Y variant of NTH1 binds oxidized DNA, but does not remove the damaged bases. Instead it blocks DNA replication and induces chromosome damage and cancer phenotype [129]. Similarly, the catalytically inactive G83D variant of NEIL has also been shown to induce genomic instability and cellular transformation [130]. 

The intense use of alternative splicing also contributes to the production of multiple isoforms of DNA glycosylases [131]. DNA glycosylase genes contain between 6 (for *NTH1*) and 17 exons (for *MUTYH*) capable of generating multiple splice variants. Alternative splicing gives rise, for example, to two groups of OGG1 isoforms by mutual exclusion of either exon 7 or 8; within the two groups, additional alternative splicing takes place to produce multiple isoforms. One function of the alternative spicing is to produce isoforms dedicated to the repair of genomic DNA and others for the repair of mitochondrial DNA. This is the case for the two isoforms of UNG, since only UNG1 possesses a mitochondrial targeting signal in its 3′ non-coding sequence [132]. In the case of human NTH1, only one out of the three known isoforms contains the catalytic domain of the protein and localizes to both the nucleus and mitochondria [131]. Several isoforms of NEIL1 and NEIL2 proteins are also missing the catalytic domains and some additionally lack the C-terminal DNA-binding domain [133]. Further studies are needed to better understand the precise roles of these different variants in both healthy and cancer cells [134,135].

DNA glycosylases also show diverse tissue-specific expression patterns [53] and are differentially expressed along the cell cycle. For example, human OGG1 is expressed at similar levels in many tissues [136], but is overexpressed in germinal centers of B cells [137]. Human NTH1 is also ubiquitously expressed with high expression in the heart and brain [138]. The expression profile of NEIL3 is particular in that it is expressed during embryonic development in stem cells and blood-forming organs [117]. With regards to the cell cycle, NTH1, MUTYH, and NEIL1 are preferentially produced in the S phase, whereas OGG1 has been shown to be expressed either in the S phase or throughout the cell cycle [99,139]. Differential expression of UNG2 and TDG in different phases of the cell cycle (UNG2 in S-phase and TDG in G2) provides a mechanism for functional separation of these two partially redundant uracil DNA glycosylases [77]. Analysis of mRNA expression patterns in various cancers (www.cBioPortal.org; [140]) reveals that several DNA glycosylases undergo amplification or upregulation in cancer cells and, as illustrated in Figure 2, this phenomenon is highly correlated with copy number variations caused by duplication, deletion, or inversion of DNA glycosylase genes in tumor cells, but can also be due to altered transcriptional regulation [8,141]. Increased levels of DNA glycosylases in cancer cells could be a response to the increased abundance of DNA lesions, but may also directly contribute to tumorigenesis through the formation of toxic repair intermediates. UNG and SMUG1 are overexpressed in non-small cell lung cancer and there is also evidence that SMUG1 is highly expressed in gastric cancer and this is correlated with poor survival [142] (Figure 2A,B). NTH1 and MPG are overexpressed in colorectal adenocarcinoma and undergo amplification in invasive breast carcinoma cells (Figure 2C,E). The allelic loss of the *OGG1* gene correlates with increased 8-oxo-G accumulation, leading to enhanced risk of oxidative stress-induced carcinogenesis in esophageal cancer (Figure 2D) [143]. NEIL3 is overexpressed in hepatocellular carcinoma, invasive breast carcinoma, and non-small cell lung cancer (Figure 2F) and this overexpression, which is independent of gene amplification, has been shown to significantly increase genomic alteration in cancer patients and to correlate with poor survival [144]. 

Two additional layers of regulation have been particularly studied over the past years, i.e., post-translational modifications (PTMs; Table 1) and protein–protein interactions (PPIs). Several reports now clearly shed light on the crucial role of PTMs as actors of DNA repair regulation and coordination [145,146]. PTMs have indeed been shown to regulate DNA glycosylase protein levels, catalytic activities, interactions with protein partners or with DNA, and dysregulation of PTMs can also lead to major changes in the DNA repair profiles of cancer cells [145,146]. Phosphorylation has so far been the most studied PTM of DNA glycosylases. For instance, the cell cycle-regulated DNA glycosylase, UNG2, is phosphorylated in a stepwise manner by early and late cyclin-dependent kinases on S23, T60, and S64 located in its N-terminal non-catalytic extremity, leading to increased catalytic activity [75]. The N-terminal domain of UNG2 contains the interaction sites for replication protein A (RPA) and proliferating cell nuclear antigen (PCNA), both of which are associated with the replicative machinery. Phosphorylation of S23 leads to a slightly increased interaction with RPA, whereas further phosphorylation at positions T60 and S64, occurring during the S phase, in contrast significantly reduces this interaction. Phosphorylation of UNG2 has also been reported to affect its interaction with PCNA. Phosphorylation of T6 and K8, located in the PCNA binding site of UNG2, has indeed been shown to reduce the interaction between the two proteins [147]. Thus, the phosphorylation state of UNG2 may finely regulate its network of interactions with partners at replicative foci. 

In the case of MPG, its phosphorylation by ATM is associated with increased resistance of pediatric cancers to the alkylating agent, temozolomide, commonly used in chemotherapy, which suggests that phosphorylation of MPG may enhance its repair activity [108]. In the case of OGG1, serine-threonine phosphorylation by the cyclin-dependent kinase CDK4 has been shown to increase its 8-oxo-G repair activity, whereas tyrosine phosphorylation of OGG1 by c-Abl does not affect its catalytic activity, illustrating here again the differential roles of phosphorylation on DNA glycosylase functions [98]. TDG, like OGG1, can be phosphorylated in vitro by protein kinase C (PKC) [87,94]; interestingly, the phosphorylation sites on TDG are very close to lysine residues acetylated by CBP/p300. Together, these two PTMs fine tune the DNA repair activity of TDG: Acetylation inhibits its catalytic activity, while phosphorylation prevents acetylation of the neighboring lysine residues, thus promoting its catalytic activity [85,87]. Acetylation of NEIL2 at a specific lysine has also be reported to inhibit both its glycosylase and AP lyase activities [114]. 

Ubiquitination and sumoylation also play a major role in regulating DDR, by determining the fate of modified proteins, including DNA glycosylases, and dysregulation of key ubiquitin E3 ligases, deubiquitinases, SUMOylases, and deSUMOylases is observed in cancer [148]. For example, the steady state level of UNG2 is regulated by successive phosphorylations followed by ubiquitination, which together control the degradation of UNG2 through the formation of a phosphodegron [75,76,77]. Ubiquitination is also largely associated with the degradation of TDG during the S phase [88,89]. MUTYH is regulated by ubiquitination and a physical interaction has been detected between this DNA glycosylase and its E3 ubiquitin ligase [104]. In this case, the ubiquitination of MUTYH diminishes its fixation to chromatin. Several examples of sumoylation of DNA glycosylases have been reported. Human TDG has been shown to be sumoylated, and this SUMO conjugation significantly increases its enzymatic turnover on G:U-containing substrates and impairs its ability to process G:T substrates [83]. A more recent study also suggests that sumoylation of TDG may facilitate the recruitment of other partner proteins to the sites of DNA damage [149]. Interestingly, overexpression of EGFP-SUMO-1 in HepG2 liver cancer cells causes a marked increase in the abundance of UDG most likely through changes at the translational level or protection from degradation [150].

PPIs constitute a complex network and take part in many cellular processes, and the identification of interacting partners can shed light on a protein’s cellular function and can also provide new therapeutic strategies in the case of disease. DNA glycosylases notably interact with other DNA repair proteins, including other BER enzymes, and the replication machinery. OGG1 for example has been reported to interact with PARP1 [151], and several DNA glycosylases including UNG2, MPG, NTH1, and NEIL2 have been reported to interact with XRCC1, PCNA, and RPA, thereby contributing to a better coordination of this multi-step repair process [152,153]. DNA glycosylases also interact with proteins belonging to other DNA repair pathways. The NER pathway is functionally very close to the BER as it repairs base damages and it induces incisions on both sides of the lesion thereby generating single-strand breaks (SSBs) [25]. Cross-talk between the two pathways has been identified. In particular, the NER enzyme Xeroderma Pigmentosum Complementation Group G (XPG) binds directly to NTH1 and enhances its binding to damaged DNA. The stimulation of NTH1’s repair activity by XPG does not require the XPG’s catalytic activity however [154,155]. A direct interaction has also been reported between Cockayne syndrome group B protein (CSB) and NEIL2, leading to a stimulated incision activity of NEIL2. Moreover, after incubation of HeLa cells with alpha amanitin, an inhibitor of transcription, CBS, and NEIL2 have been shown to be recruited to the site of the stalling replication fork, where the interaction then occurs [156,157]. A functional link between BER and DSB repair has also been demonstrated. Kiraly and colleagues have shown using *MPG* null mice that intermediates of the BER pathway stimulate the activation of HR [158]. This is because, if unrepaired by the BER pathway, SSBs can indeed be converted into double-strand breaks (DSBs) when two SSBs occur in close proximity on opposite strands of the DNA, or if the replication or transcription machineries process SSBs [46]. Moreover, a direct interaction between the two main isoforms of OGG1 and the HR Rad52 protein has been detected both in vitro and in cellulo [159]. 

Other protein partners of DNA glycosylases have also been described, one of which is an established metastatic marker [160,161], the Y-box binding protein 1 (YB1). A direct interaction between NTH1 and the multifunctional YB1 protein has indeed been reported to occur in tumor cells in response to genotoxic stress. Through this interaction, YB1 has been shown to stimulate the AP-lyase activity of NTH1 [106,162]. Several reports have demonstrated that the level of nuclear expression of YB1 is predictive of drug resistance and poor prognosis [163,164,165,166,167,168] and the increased abundance of the NTH1-YB1 complex in tumor cells after exposure to genotoxic stress, including cisplatin treatment, could explain the increased resistance of these cells to anticancer agents [169,170]. 

## 4. Modulating DNA Glycosylase Levels as a Means of Studying Their Implication in Cancer Development

Because of their critical role in the repair of base lesions, DNA glycosylases have been and still are the focus of many researchers worldwide, wishing to determine the role of these enzymes in tumorigenesis and cancer development, but also in anticancer drug resistance. In many cases, cancer cells become resistant to initially effective DNA-damaging agents [24,25,26], and overexpression of DNA glycosylases is often observed in such cells [144,171,172]. To evaluate the contribution of individual DNA glycosylases to tumorigenesis, researchers have either up- or down-regulated the expression of DNA glycosylases from all four superfamilies in tumor cells. The knockdown of DNA glycosylase genes in particular has been a powerful method to study the physiological importance of these enzymes [59,68]. Several examples of such studies are presented below, which together illustrate the complex and multivalent roles of DNA glycosylases in cancer [173].

One such study demonstrated that homozygous mutant ES mice cells in which *MPG* was knocked out were more sensitive than wild-type cells to different types of alkylating agents, including methyl methanesulfonate (MMS) that produces different types of lesions and MeOSO_2_ (CH_2_)_2_-lexitropsin that produces almost exclusively 3meA lesions [174,175]. In contrast, in another study, decreased expression of MPG in *MPG*-deficient bone marrow mouse cells confers instead resistance to alkylating agents [66]. Moreover, overexpression of MPG in a human mammary gland adenocarcinoma cell line, sensitizes the cells to treatment with different alkylating agents (MMS, N-methyl-NV-nitro-N-nitroso- guanidine, methyl nitrosourea, dimethyl sulphate, and temozolomide) [176]. These apparently contradictory results have since been explained by the finding that the cytotoxic BER intermediates, including 5′ dRP lesions, are efficiently repaired in certain cell lines, but not in others [177].

Studies of the HhH DNA glycosylase superfamily showed that *OGG1* null mice are viable, but accumulate more 8-oxo-G DNA damage than wild-type mice [178]. Moreover, Yang and colleagues showed that knock down of the two HhH DNA glycosylases, *OGG1* and *NTH1*, in TK6 human B lymphoblast cells sensitizes cells to hydrogen peroxide (H_2_O_2_) treatment, whereas the overproduction of these two DNA glycosylases on the contrary protects the cells from H_2_O_2_ damage [179]. This was confirmed by Preston et. al., who demonstrated that overexpression of OGG1 alone in human embryonic kidney cells results in increased repair of 8-oxo-G induced by H_2_O_2_ or cisplatin [180]. However, the difference in sensitivity between wild-type mice MEF cells and *NTH1* null mice MEF cells treated with H_2_O_2_ is not statistically significant [179], probably because of the partial redundancy in the activity and substrate specificity of DNA glycosylases, which has been clearly demonstrated for the NEIL superfamily [181]. The redundancy in the glycosylase function in vivo is further supported by the fact that knocking down *NTH1* alone did not cause significant consequences in mice [182], whereas targeted deletion of both *NTH1* and *NEIL1* resulted in increased tumorigenesis in lung and liver [183]. This study also demonstrated for the first time the carcinogenic nature of oxidative base lesions other than 8-oxo-G, and highlighted the importance of NTH1 in the repair of such lesions [183]. Interestingly, a recent study has shown that overexpression of NTH1 in non-transformed human bronchial epithelial cells is responsible for genomic instability, loss of contact inhibition, and growth in soft agar, proving that NTH1 overexpression can also contribute to cell transformation [172]. 

Taken together, these studies reveal the key, but complex roles of DNA glycosylases in cancer [173]. Depending on the enzyme, the availability of downstream enzymes, the cell type, the cellular environment, and the status of key regulators such as ATM, ATR, or p53, overexpression of DNA glycosylases can either favor cell transformation, tumorigenesis, and/or drug resistance, or instead sensitize cells to oxidative damage by enhancing the production of toxic BER products [20]. A better understanding of the molecular mechanisms that regulate the levels and activity of DNA glycosylases in tumors is thus a key step to drive the development of new anticancer drugs targeting these enzymes.

## 5. Finding Inhibitors of DNA Glycosylases

Inhibiting DNA glycosylases is now considered a valid strategy to eliminate cancer cells. However, because of the partial redundancy of these enzymes and the cross-talk between different repair pathways, the efficiency of DNA glycosylase inhibitors as monotherapy is expected to be limited, but should be greatly enhanced when used either in combination therapy together with a conventional DNA-damaging chemotherapeutic agent, or in tumors displaying defects in alternative repair pathways (synthetic lethality concept) in a more personalized medicine approach. Currently, most efforts have aimed to identify small molecule inhibitors of the catalytic activities of DNA glycosylases. Different approaches have been used with success: (i) Targeted low-throughput approaches, (ii) computational- and structure-based rational drug design, and (iii) high-throughput screening (HTS) of chemical or fragment-based libraries, all of which have recently been shown to be valid and complementary strategies to find potent inhibitors of DNA glycosylases [184]. 

Several metabolic cofactors have been shown to efficiently inhibit DNA glycosylases. This is the case of pyridoxal 5′-phosphate (PLP), a cofactor of enzymes involved in amino acid metabolism. The aldehyde moiety of PLP has been shown to inhibit or inactivate diverse DNA-dependent enzymes and was therefore tested on several DNA glycosylases. Of these, only NEIL2 was significantly inhibited by PLP due to the formation of a Schiff base between PLP and a DNA-binding loop in the enzyme [185]. Similarly, the intermediate of tyrosine catabolism, fumarylacetoacetate (FAA), was also found to specifically inhibit a subset of DNA glycosylases. In particular, the NEIL1 and NEIL2 enzymes were strongly inhibited, whereas only a small effect was observed in vitro on NTH1 and OGG1 and no effect on UNG2 [186]. FAA inhibition of DNA glycosylases may explain the increased mutagenesis rates associated with hepatocarcinoma development in HT1 patients, which have a deficiency in FAA hydrolase and thus accumulate high intracellular levels of this intermediate catabolite. 

Given the availability of high-resolution crystallographic structures of several human DNA glycosylases (UNG, TDG, OGG1, NEIL1, NEIL3, MUTYH, MPG, and MBD4), computational modeling and structure-based drug design have also been used to identify and optimize DNA glycosylase inhibitors. A recent computational druggability assessment study revealed that DNA glycosylases are druggable targets, with OGG1, MUTYH, NEIL1, UNG, and TDG being the most favorable drug-binding proteins [184]. Structural studies of bacterial Fpg and human NEIL enzymes are currently been exploited, for example, to develop NEIL1 inhibitors derived from the 2-thioxanthine (2TX) compound that was originally found to specifically inhibit bacterial Fpg and not its human homolog [187,188]. Structure-based protein engineering has been used to improve the selectivity of SAUGI, an inhibitory protein from *Staphylococcus aureus*, for human UDG versus Herpes simplex virus (HSV) UDG by comparing the crystal structures of SAUGI-human UDG with that of SAUGI-HSV-UDG [189]. Structural studies have also guided the optimization of fragment-based inhibitors of human UDG [190]. 

To identify UNG inhibitors, Jiang and colleagues developed a uracil-directed ligand tethering strategy, in which a uracil-aldehyde ligand was tethered via alkyloxyamine linker chemistry to a diverse array of aldehyde binding elements. The goal was to exploit the uracil ligand to target the UNG active site and the alkyloxyamine linker tethering to randomly explore peripheral binding pockets. This original approach rapidly identified the first small molecule inhibitors of human UNG with micromolar to submicromolar binding affinities [191] and the best inhibitor was co-crystallized with the UNG2 enzyme. The structure of inhibitor-bound UNG2 revealed that the inhibitor engages in crucial electrostatic interactions and hydrogen bonding with the enzyme, similar to those seen when complexed to uracil-containing DNA [192]. 

Over the past decade, novel fluorescence-based HTS approaches have been developed to identify DNA glycosylase inhibitors. These assays respond to three essential criteria: Fast, robust, and adaptable (Figure 3). The use of such approaches has led to the selection and validation of several efficient drugs against DNA glycosylases. In a recent and precursor study, Jacobs et al. developed a fluorescence-based assay adapted for HTS to find inhibitors of NEIL1 [193], which has been identified as a possible candidate for a synthetic lethality approach in Fanconi anemia disease [194]. This first study was inspired by an HTS for inhibitors of APE1 [195]. It is designed to select inhibitors of bifunctional DNA glycosylases, as it is based on strand incision activity that occurs after hydrolysis of the glycosylic bond. In this assay, a short synthetic oligodeoxyribonucleotide containing secondary oxidation products of 8-oxo-G (spirodihydantoin, Sp, and guanidinohydantoin, Gh) and labeled at its 5′ end with the TAMRA red fluorophore was hybridized to a complementary DNA containing a quencher molecule at its 3′ end (Figure 3A). Upon repair of the site-specific lesion by NEIL1, the DNA fragment containing the TAMRA is incised and released from the DNA duplex, leading to increased fluorescence. The relative fluorescence can then be measured both in the presence and absence of the inhibitor. HTS led to the selection of purine analogs with IC_50_ values ranging from 4 to 25 μM. Interestingly, four of them significantly inhibited the in vitro repair activity of NEIL1 on γ-irradiated calf thymus DNA. Although the selected inhibitors were also shown to significantly block the activity of closely-related DNA glycosylases such as NTH1, this study established the first customized fluorescence-based assay for HTS to find inhibitors of DNA glycosylases. 

As OGG1 is an essential bifunctional DNA glycosylase responsible for removing the most abundant oxidized base produced in cells, it is now considered as a candidate of choice for the development of anticancer drugs. To do this, the fluorescence-based assay developed by Jacobs and colleagues was adapted to OGG1 [196]. The authors identified hydrazide or acyl hydrazine-based inhibitors that display submicromolar IC_50_ values against OGG1 incision strand activity. These very encouraging results were further confirmed by conventional gel shift assays, confirming the potential of such HTS using miniaturized assays to find small molecule inhibitors of DNA glycosylases. Moreover, the selected inhibitors were found to be specific to OGG1; little to no inhibition of NEIL1, NTH1, and bacterial Fpg assessed on their respective substrates (FapyG for NEIL1 and NTH1 and 8-oxo-G for Fpg) was detected. Further structural analyses of inhibitor-bound OGG1 will be needed to shed light on the mode of inhibition of these promising molecules and the molecular contacts underlying this unexpected specificity. 

The assays described above, however, are not suitable for measuring the monofunctional glycosylase activity of DNA glycosylases since they rely on the release of the fluorophore-labeled strand after strand incision by the AP lyase activity. In vivo, OGG1, like NTH1, has been shown to act predominantly as a monofunctional DNA glycosylase [198]; thus, the assay presented in Figure 3A may not be the most adapted to find potent OGG1 inhibitors. Different technical solutions have been found to select inhibitors of the DNA glycosylase activity. Mancuso and colleagues added APE1 to their molecular beacon-based assay to detect the monofunctional glycosylase activity of TDG [199]. They screened 2000 drugs and identified 20 candidate inhibitors. Some of these compounds, like juglone and closantel, were confirmed as TDG inhibitors in a standard DNA glycosylase assay. These compounds also led to a dose-dependent reduction in cell viability of melanoma cells with IC_50_ values close to 10 µM, which suggests that interfering with the DNA repair and epigenetic activity of TDG may represent a new and valid approach for the treatment of melanoma. A similar strategy was also used recently in a search for new therapeutic strategies to fight inflammation. Visnes and colleagues selected a site-specific inhibitor of OGG1, named TH5487, which blocks OGG1’s ability to recognize and repair 8-oxo-G containing DNA [200]. A recent crystal structure of human OGG1 bound to this inhibitor (unpublished; PDB 6RLW) reveals the intricate interactions formed between the inhibitor and the substrate binding pocket of OGG1 (Figure 4A,B). 

Tahara and colleagues used a different strategy to target the glycosylase activity of OGG1. They adapted a previously reported fluorogenic assay [203] to study the release of 8-oxo-G from DNA [197]. This assay makes use of a fluorescent hairpin oligomer probe, called OGR1 [204], containing a modified 8-oxo-G base linked to a quencher that specifically quenches a highly fluorescent DNA base analogue, tCo, covalently bound to the base of the neighboring nucleotide (Figure 3B). After cleavage and release of the 8-oxo-G moiety and its associated quencher, the tCo molecule on the oligonucleotide becomes fluorescent and the fluorescent signal can be followed in real time. In this assay, only the DNA glycosylase activity is measured, allowing us to specifically screen and select drugs that interfere with the break-down of the N-glycosidic link between the 8-oxo-G base and the deoxyribose. After HTS, one compound named SU0268, which has an acyl tetrahydroquinoline sulfonamide skeleton with an IC_50_ of 59 nM was selected (Figure 4C). This competitive inhibitor is highly specific to OGG1, shows no cytotoxicity in cells, and inhibits OGG1 activity both in vitro and in cells, where the abundance of 8-oxo-G bases in cells incubated with 0.5 µM SU0268 increased to levels comparable to those detected in cells treated with 0.5 H_2_O_2_ and 0.3 mM of Cr^3+^ [197].

Upon oxidative stress, dGTP is in part transformed into 8-oxo-dGTP that can be incorporated into DNA. To prevent this, a nudix hydrolase, the MutT homolog-1 (MTH1) enzyme converts 8-oxo-dGTP into 8-oxo-dGMP, which can no longer be incorporated into DNA [205]. Recently, MTH1 was also identified as an interesting target for inhibitor screening in the context of cancer, and several small molecule drugs were identified [206]. Based on these studies, Tahara and colleagues set out to design a dual inhibitor corresponding to the association of two molecules, one specific for MTH1 and the other being SU0268 (the inhibitor of OGG1), giving rise to compound SU0383 (Figure 4D). This small molecule inhibits OGG1 with an IC_50_ of 49 nM, measured using the OGR1 probe. It also inhibits MTH1 with an IC_50_ of 34 nM, using a luminescence-based assay for the activity of MTH1 in the conversion of 8-oxo-dGTP to 8-oxo-dGMP [207]. Interestingly, exposure of MCF7 breast cancer cells to 16 µM of H_2_O_2_ in the presence of SU0383 leads to a drop in cell viability of 20%, demonstrating the ability of the molecule to increase the sensitivity of the cells towards oxidative stress [202]. 

In a recent study conducted by Senarisoy et al., an alternative approach was proposed to restore sensitivity of drug-resistant tumor cells, by targeting the interaction of NTH1 with one of its cellular partners, YB1, instead of targeting the catalytic activities of NTH1 [162]. YB1 is a transcription factor that has been shown to bind to NTH1 and stimulate its AP-lyase activity. In drug-resistant tumor cells, nuclear localization of YB1 is elevated and this favors the interaction of NTH1 with YB1, leading to an increased abundance of the NTH1-YB1 complex. In this study, inhibitors of the NTH1-YB1 interaction were identified and validated using a FRET-based biosensor that was designed for HTS (Figure 3C). In this biosensor construct, the fluorescent protein, sYFP2, was placed at the amino terminus followed by YB1 and NTH1, and a second FRET-compatible fluorescent protein, mTQ2, at the C-terminus. Using this construct, HTS of 1200 molecules was performed and 8 potent inhibitor molecules were selected. The inhibitory effects of these molecules were further confirmed by AlphaLISA and their molecular targets were identified using thermal shift assay. Of the 8 inhibitors, two molecules, meclocycline and oxytetracycline, were found to have low IC_50_ values, i.e., 1.5 and 10.3 μM. To further investigate their inhibitory properties in drug-resistant tumor cells, cisplatin-resistant MCF7 cells were used and the effects of the inhibitor molecules on the sensitivity of MCF7 cells to cisplatin was studied. The results of this study show that both meclocycline and oxytetracycline induced a small, but significant, concentration-dependent decrease in the viability of MCF7 cells treated with cisplatin, indicating that these molecules partially restore the sensitivity of resistant MCF7 cells to cisplatin [162]. This study thus demonstrates that the PPIs involving DNA glycosylases also constitute druggable targets for the development of new therapeutic strategies to eliminate cancer cells.

## 6. Conclusions

Because of the central role of BER in the maintenance and repair of DNA damage following cancer treatment, BER enzymes have been the focus of numerous studies over the past two decades, leading to the development of several potent drugs that are now in clinical use [25,38]. Despite these successes, DNA glycosylases that initiate the BER pathway have received much less attention and the development of specific inhibitors of these enzymes has been lagging. Yet, it is now clear that dysregulation of DNA glycosylases plays a central role in numerous cancers and at different stages of the disease, favoring both the onset (cell transformation), but also later stages such as metastasis [43,44,45]. Recently, DNA glycosylases such as NEIL1, OGG1, and NTH1 have been identified as potential targets for combination chemotherapeutic strategies, and several HTS assays have been developed to screen chemical libraries to select small molecule inhibitors of the catalytic activities of DNA glycosylases [162,193,197,202]. Both highly specific and broad-spectrum inhibitors have been successfully identified with such screens and initial tests on tumor cells are very promising, with several compounds capable of increasing the sensitivity of tumor cells to cytotoxic agents. It should be noted, however, that in several of the studies described above [162,193,197,202], simple cell-based assays measuring cell viability were used to evaluate the potential anti-cancer effects of the selected drugs. Although these types of assays give insightful preliminary results, they have, nonetheless, a number of limitations (recently reviewed in [208]). For this reason, before envisaging clinical trials, a crucial step is now to conduct more advanced preclinical assays to validate these findings. These typically include the use of animal models, but also patient-derived xenografts, which are compatible with HTS [209], spheroids [210], organoids [211] in combination or not with CRISPR-Cas9 technology [212], or organ-on-chip technology [213], the aim of which is to mimic the reconstitution of tissues (normal or tumoral) in their 3D structure and in their microenvironment. 

Because inhibiting DNA glycosylase activity in healthy cells could have dramatic consequences, alternative drug targets have also been identified that can modulate the activity of DNA glycosylases more specifically in tumor cells. Targeting PPIs between DNA glycosylases and their cellular partners that are enhanced in tumors, as was done for the NTH1-YB1 interaction [162], constitutes a new and original approach, which has the advantage of not blocking DNA repair in normal, healthy cells, but instead to specifically inhibit the upregulation of DNA repair activity in drug-resistant tumor cells. DNA glycosylases are tightly regulated at the gene, mRNA, and protein levels, and in the future each one of these regulatory systems may represent a potent drug target to fine tune the activity of specific DNA glycosylases in a more personalized medicine approach.

Beyond their potential use in anticancer therapy, drugs targeting DNA glycosylases would also have numerous applications in fundamental research to decipher the complex regulatory networks underlying the coordinated action of the various DNA repair pathways in both normal tissue and in the context of cancer.

## Figures and Tables

**Figure 1 ijms-21-09226-f001:**
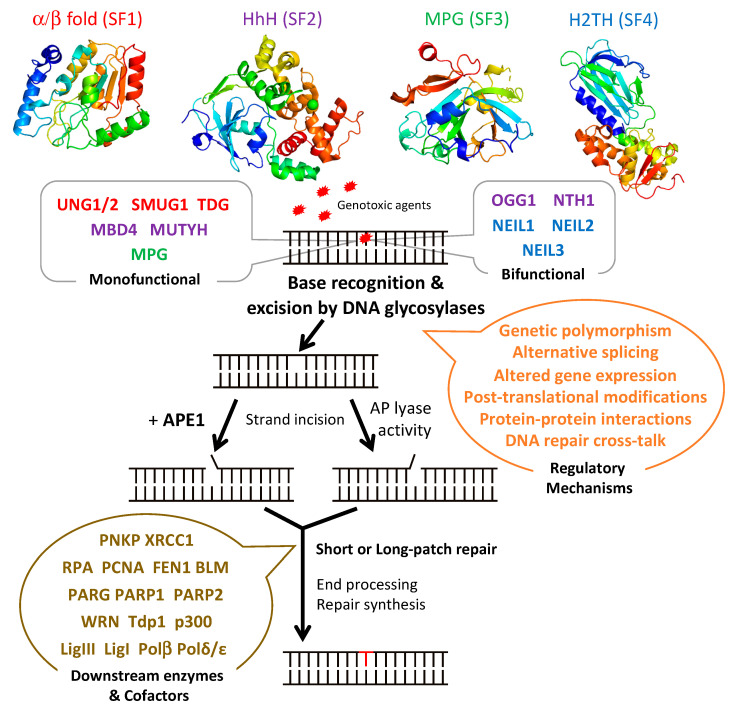
Schematic diagram illustrating the steps and enzymes involved in the base excision repair (BER) pathway. Representative structures of the different superfamilies (SF) of DNA glycosylases (SF1, α/β fold family, red; SF2, helix-hairpin-helix (HhH) family, purple; SF3, 3-methyl-purine glycosylase (MPG) family, green; SF4, helix-two-turn-helix (H2TH) family, blue) responsible for recognition and removal of damaged bases are shown. After cleavage of the damaged strand by an apurinic/apyrimidinic (AP) endonuclease, APE1, or by the AP lyase activity of bifunctional DNA glycosylases, downstream BER enzymes together with several cofactors (listed in brown) prepare the damaged site for de novo synthesis using one of two sub-pathways: short-patch or long-patch repair. DNA glycosylases are tightly regulated at the gene, mRNA, and protein levels by a set of regulatory systems (listed in orange). UNG1/2: uracil-N glycosylase 1 or 2; SMUG1: single-strand-specific monofunctional uracil DNA glycosylase 1; TDG: thymine DNA glycosylase; MBD4: methyl-CpG-binding protein 4; MUTYH: MutY homolog DNA glycosylase; OGG1: 8-oxo-G DNA glycosylase 1; NTH1: endonuclease III-like 1; NEIL1-3: endonuclease VIII-like 1-3.

**Figure 2 ijms-21-09226-f002:**
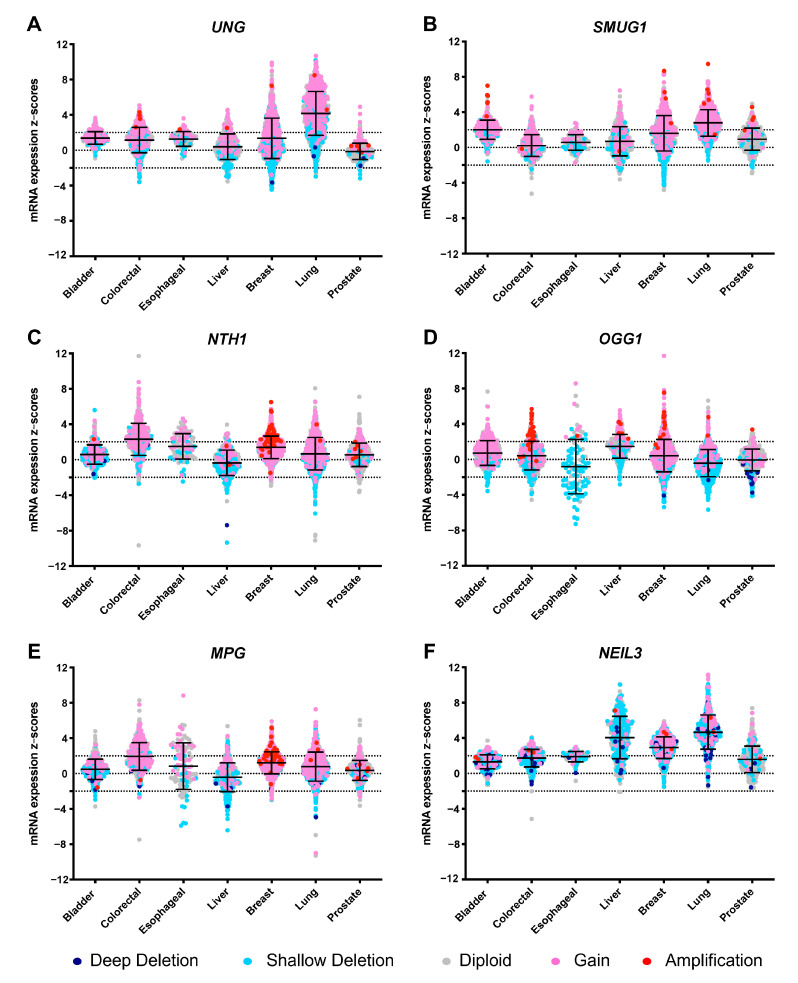
mRNA expression profiles of six representative DNA glycosylase genes in various carcinomas. mRNA expression profiles of *UNG* (**A**), *SMUG1* (**B**), *NTH1* (**C**), *OGG1* (**D**), *MPG* (**E**) and *NEIL3* (**F**). Data were extracted from the cBio Cancer Genomics Portal [140] by computing the expression of an individual gene in a tumor sample (z-score) relative to the gene’s expression in the normal sample. Samples with expression z-scores >2 or <−2 (indicated with dotted lines) in any queried gene are considered altered. mRNA expression levels are organized according to the copy-number level per gene: Deep deletion (dark blue) indicates a deep loss, possibly a homozygous deletion; shallow deletion (sky blue) indicates a shallow loss, possibly a heterozygous deletion; diploid (gray); gain (pink) indicates a low-level gain (a few additional copies, often broad); and amplification (red) indicates a high-level amplification (more copies, often focal). Cancer types: Bladder urothelial carcinoma (bladder; N = 404), colorectal adenocarcinoma (colorectal; N = 590), esophageal squamous cell carcinoma (esophageal; N = 94), hepatocellular carcinoma (liver; N = 358), invasive breast carcinoma (breast; N = 1068), non-small cell lung cancer (lung; N = 991), and prostate adenocarcinoma (prostate; N = 488).

**Figure 3 ijms-21-09226-f003:**
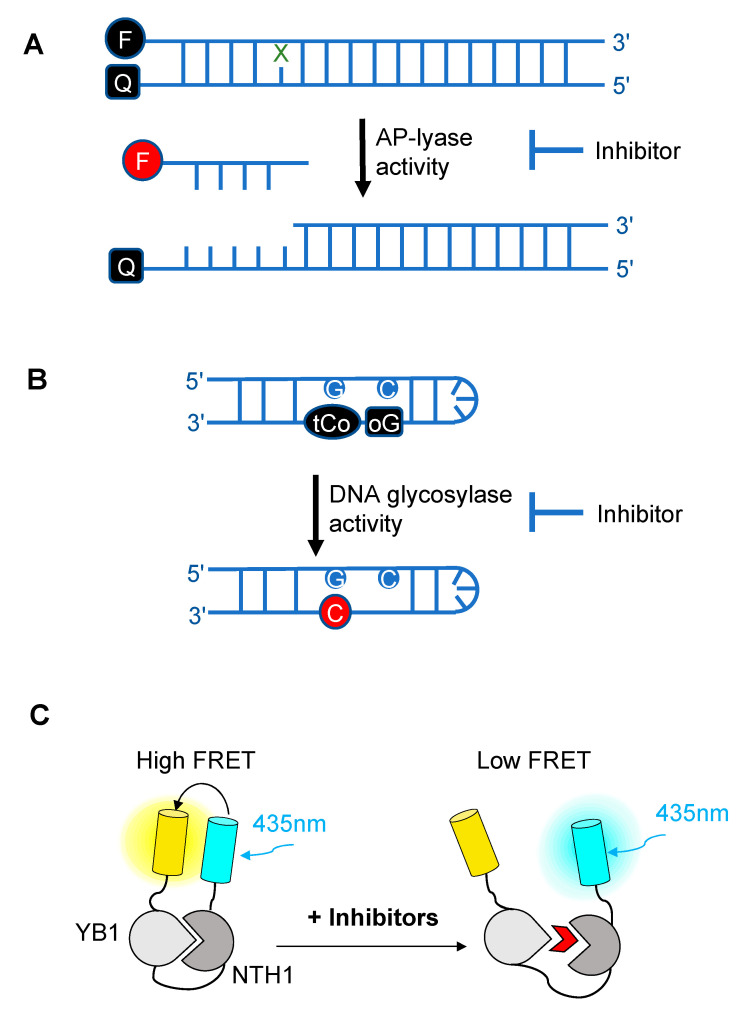
Schematic diagrams illustrating the different fluorescence-based assays developed over the past decade for high-throughput screening (HTS) of chemical libraries for the selection of inhibitors of either the AP-lyase activity [193] (**A**), the DNA glycosylase activity [197] (**B**) of DNA glycosylases, or of the interaction interface (**C**) between a DNA glycosylase (here, NTH1) and its cellular partner (YB1) [162]. FRET: Förster resonance energy transfer. In (**A**), X denotes a damaged base processed by DNA glycosylases. Cleavage by the AP lyase activity results in the release of the fluorophore-labeled lesion-containing strand and fluorescence emission (red F). In (**B**), release of a modified 8-oxo-G base (oG) linked to a quencher that specifically quenches the highly fluorescent DNA base analogue, tCo, covalently bound to the neighboring base, leads to fluorescence emission (red C). In (**C**), NTH1-YB1 complex formation is associated with high FRET levels, which are significantly reduced by inhibitors (red wedge) of the PPI interface.

**Figure 4 ijms-21-09226-f004:**
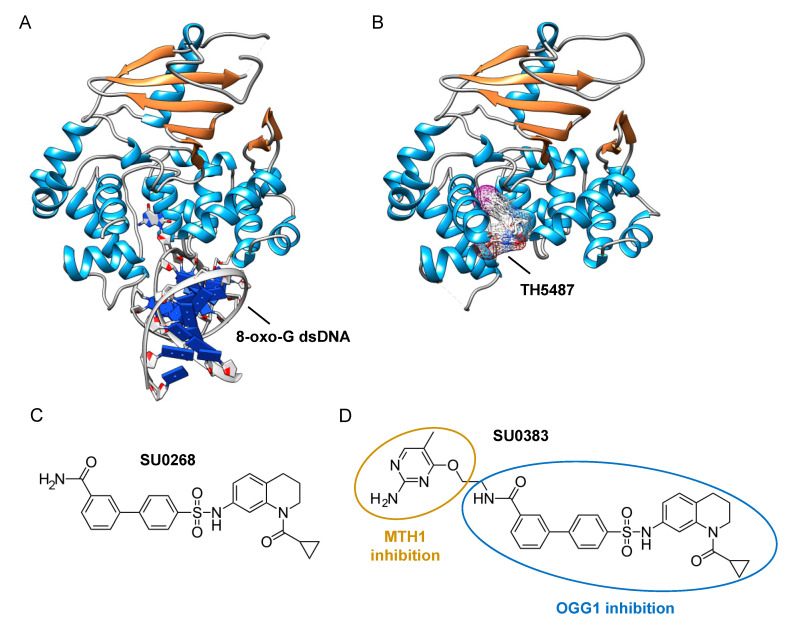
Identification of specific inhibitors of human OGG1. (**A**) Crystal structure of human OGG1 crosslinked to 8-oxo-G containing dsDNA (PDB: 6W0M; [201]). (**B**) Crystal structure of human OGG1 bound to an active site-specific inhibitor, TH5487 (PDB 6RLW), represented in ball and sticks and as a transparent mesh. (**C**) Chemical structure of the highly specific OGG1 inhibitor selected by Tahara and colleagues [197], SU0268. (**D**) Chemical structure of the dual inhibitor, SU0383, which efficiently blocks both MTH1 and OGG1 activities [202].

**Table 1 ijms-21-09226-t001:** Characteristics of human DNA glycosylases.

Structural Motif Superfamily	Name of the Glycosylase		Mono/Bifunc	Major Substrate Specificity	Reported PTMs	Ref.
Alpha-beta fold (UDG) superfamily	Uracil-N glycosylase1	UNG1	Mono	U in any context, in ss and dsDNA		[69,70,71]
	Uracil-N glycosylase2	UNG2	Mono	Similar to UNG1	Phospho Ubiquit Acetyl	[71,72,73,74,75,76,77,78,79]
	Single-strand-specific monofunctional uracil DNA glycosylase 1	SMUG1	Mono	ssU, U:G, U:A, 5-hydroxymethylU, in ss and dsDNA	Ubiquit	[72,80]
	Thymine DNA glycosylase	TDG	Mono	U:G, T:G, oxidized/deaminated 5-methylC:G, in dsDNA	Phospho Ubiquit Sumoyl Acetyl	[81,82,83,84,85,86,87,88,89]
Helix-hairpin-helix (HhH) superfamily	Methyl-binding domain glycosylase 4	MBD4	Mono	U:G and T:G, 5-hydroxymethylU in CpG islands, in dsDNA	Phospho	[90,91,92]
	8-OxoG DNA glycosylase 1	OGG1	Bifunc	Oxidized purines (8-oxoG:C, FapyG:C), in dsDNA	Phospho Nitrosyl Acetyl Ubiquit	[93,94,95,96,97,98,99]
	MutY homolog DNA glycosylase	MUTYH	Mono	A opposite 8-oxo-G/C/G, in dsDNA	Phospho Ubiquit	[100,101,102,103,104]
	Endonuclease III-like 1	NTH1	Bifunc	Oxidized pyrimidines (Tg, 5-hydroxyC, 5-hydroxyU), FapyG, FapyA, in dsDNA	Ubiquit	[105,106,107]
3-methyl-purine glycosylase (MPG) superfamily	3-methyl-purine glycosylase	MPG	Mono	3-methylA, 1-methyl, 7-methylG, εA, ethenoA hypoxanthine, in ss and dsDNA	Phospho Acetyl	[108,109,110,111]
Hairpin-2- turn-hairpin (NEIL) superfamily	Endonuclease VIII-like glycosylase 1	NEIL1	Bifunc	Oxidized pyrimidines (Tg, 5-hydroxyU, 5,6-dihydroU, hydantoins Gh and Sp), FapyG, FapyA, in ss and dsDNA		[57,67,112,113]
	Endonuclease VIII-like glycosylase 2	NEIL2	Bifunc	Similar to NEIL1 in bubbles and loops	Acetyl	[114,115]
	Endonuclease VIII-like glycosylase 3	NEIL3	Bifunc	Similar to NEIL1 (FapyG, FapyA, Sp and Gh) in ssDNA		[41,116,117,118]

Mono: Monofunctional DNA glycosylase; Bifunc: Bifunctional DNA glycosylase; Phospho: Phosphorylation; Ubiquit: Ubiquitination; Sumoyl: Sumoylation; Acetyl: Acetylation; Nitrosyl: Nitrosylation.

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
