# Peer review of "Focus on DNA Glycosylases—A Set of Tightly Regulated Enzymes with a High Potential as Anticancer Drug Targets"

_ijms, 2020, doi:10.3390/ijms21239226_

Round 1

Reviewer 1 Report

This is another review on the DNA glycosylases as targets in cancer therapy. It is a well written; however, it needs an extensive revision as explained below.

Critique:

Line 29-32: There are many other articles dealing with the accumulation of DNA damage in cancer cells. Refs. 4-8 do not deal in general with the accumulated DNA lesions. Some of the only concentrate on 8-oxo-Gua as usually dome, although many other DNA lesions the accumulation of which has been reported in cancer cells.

Lines 46-48 – This sentence doesn’t make any sense. The meaning is not clear. It needs to be rewritten.

Lines 48-61 – The authors do not comment that the primary cytotoxic mechanism of PARP1/2 inhibitors is the formation of stable PARP-DNA complexes and that this effect significantly surpasses the inhibition of the polymerase function as related to assisting in the formation of BER complexes.

Lines 62-66 – HR and NHEJ are damage tolerance mechanisms, not repair mechanisms

Lines 81-84 – There are many other mutagenic lesions than 8-oxo-Gua (Refs. 34, 38). Tg is not a mutagenic lesion, but a lethal lesion (Ref. 38).

Lines 85-88 – The authors omitted XRCC1 as an essential scaffolding protein for much of the secondary assembly of the BER complex that is downstream of glycosylase activities. Also PNKP should be mentioned since it is essential in the repair initiated by NEIL1 and NEIL2.

Figure 1 – NEIL3 is a bi-functional glycosylase, not a monofunctional glycosylase (see Liu et al. PNAS 107, 4925, 2010, which also has much more information on the substrates, etc. of NEIL3; also see Ref. 32).

Lines 107-114 – the active site residues of NEIL1 and NEIL2 is the secondary amine of the N-terminal proline and thus bypasses the need for APE1, but requires PNKP.

Lines 123-128 – There are many more lesions as substrates of DNA glycosylases than those listed here. An extensive recent review of the substrates of DNA glycosylases can be found in Ref. 32.

Table 1 – 1: Which Fapy is meant by Fapy:C? OGG1 excises 8-oxoG and FapyG with similar excision kinetics (Ref. 32).

2: NTH1 also excises FapyA.

3: FapyG and FapyA are not the products of pyrimidines. They are produced from Gua and Ade, respectively.

4: 8-oxoG is a not a substrate of NEIL1. This has been shown many times (see e.g., Refs.  28, 94, 162. There are many other refs. to this effect).

5: Gh and Sp have been studied using oligo-2’-deoxynucleotides only, not using genomic DNA.

6: Again, NEIL3 is bi-functional, not monofunctional (see Liu et al. PNAS 107, 4925, 2010).

Lines 162-163 – The statement with Ref. 107 is not correct. The Ser326Cys OGG1 variant has reduced catalytic activity for 8-oxoG and FapyG compared to the wild-type (see NAR 27, 4001, 1999, and Ref. 32).

Lines 273-330 – This section does not describe any of the phenotypes associated with loss of NEIL1 in mice and only anecdotally mentions OGG1 knockout mice – there are significant literatures on these phenotypes, most of which are inside and outside the cancer-specific topics (see e.g., Ref. 162 for several cancer incidences in NEIL1- and NTH1-knockout mice).

Line 351 – The NEIL1 substrate for inhibitor studied was the secondary oxidation products of 8-oxoG. 

Line 368-369 – No ref. was given for this statement. Which other DNA glycosylases repair 8-oxoG besides human and other eukaryotic OGG1s? Not NEIL1, NEIL2, NEIL3, NTH1! MutY is a not a part of BER. E. coli Fpg. homolog of OGG1 does remove 8-oxoG.

In summary, this is a good review of DNA glycosylases as potential targets in cancer therapy. However, it needs an extensive revision as explained in the afore-mentioned comments before it can be considered for publication.

Author Response

This is another review on the DNA glycosylases as targets in cancer therapy. It is a well written; however, it needs an extensive revision as explained below.

  • We thank the reviewer for his/her valuable and constructive criticisms, which have substantially contributed to improving our manuscript.

Critique:

Line 29-32: There are many other articles dealing with the accumulation of DNA damage in cancer cells. Refs. 4-8 do not deal in general with the accumulated DNA lesions. Some of the only concentrate on 8-oxo-Gua as usually dome, although many other DNA lesions the accumulation of which has been reported in cancer cells.

  • We have modified the references relating to the accumulation of DNA damage in cancer cells and to the role of ROS in DNA damage. Inappropriate references 4 and 5 were removed, and additional references have been added (now references 4 to 9, line 32).

Lines 46-48 – This sentence doesn’t make any sense. The meaning is not clear. It needs to be rewritten.

  • This sentence has been rephrased. Line 46: “As a result, development of drugs that target DNA repair pathways has attracted much attention over the recent years in the cancer drug discovery field and new innovating anticancer strategies targeting DNA repair proteins have emerged”.

Lines 48-61 – The authors do not comment that the primary cytotoxic mechanism of PARP1/2 inhibitors is the formation of stable PARP-DNA complexes and that this effect significantly surpasses the inhibition of the polymerase function as related to assisting in the formation of BER complexes.

  • We believe that such a level of detail regarding the mode of action of PARP inhibitors is not needed here. The main point is to illustrate that DNA repair inhibitors have reached clinical use. As indicated by the reviewer, by blocking the catalytic activity of PARPs these inhibitors influence both PARP’s association with DNA and its polymerase function. We have modified lines 50 and 51 to add this point.

Lines 62-66 – HR and NHEJ are damage tolerance mechanisms, not repair mechanisms

  • For clarity, we prefer to use the term “repair” in all cases, which is commonly used to describe all processes aiming to restore the integrity of DNA, although we agree that the term “damage tolerance” may be more appropriate when discussing double-strand break repair.

Lines 81-84 – There are many other mutagenic lesions than 8-oxo-Gua (Refs. 34, 38). Tg is not a mutagenic lesion, but a lethal lesion (Ref. 38).

  • This has been clarified in the revised manuscript, lines 82 and 83.

Lines 85-88 – The authors omitted XRCC1 as an essential scaffolding protein for much of the secondary assembly of the BER complex that is downstream of glycosylase activities. Also PNKP should be mentioned since it is essential in the repair initiated by NEIL1 and NEIL2.

  • The description of the BER process has been extended so as to provide a more thorough account of this complex repair pathway including the two sub-pathways (short- and long-patch repair) and the key players involved in each case. See lines 92-102 and new Figure 1.

Figure 1 – NEIL3 is a bi-functional glycosylase, not a monofunctional glycosylase (see Liu et al. PNAS 107, 4925, 2010, which also has much more information on the substrates, etc. of NEIL3; also see Ref. 32).

  • This has been corrected in Table 1 and the additional reference has been added.

Lines 107-114 – the active site residues of NEIL1 and NEIL2 is the secondary amine of the N-terminal proline and thus bypasses the need for APE1, but requires PNKP.

  • This information has been added.

Lines 123-128 – There are many more lesions as substrates of DNA glycosylases than those listed here. An extensive recent review of the substrates of DNA glycosylases can be found in Ref. 32. Table 1 – 1: Which Fapy is meant by Fapy:C? OGG1 excises 8-oxoG and FapyG with similar excision kinetics (Ref. 32). 2: NTH1 also excises FapyA. 3: FapyG and FapyA are not the products of pyrimidines. They are produced from Gua and Ade, respectively. 4: 8-oxoG is a not a substrate of NEIL1. This has been shown many times (see e.g., Refs.  28, 94, 162. There are many other refs. to this effect). 5: Gh and Sp have been studied using oligo-2’-deoxynucleotides only, not using genomic DNA. 6: Again, NEIL3 is bi-functional, not monofunctional (see Liu et al. PNAS 107, 4925, 2010).

  • We agree with the reviewer that there are many more substrates than those listed in Table 1. Our objective was to provide an overview of the major substrates of each DNA glycosylase as stated in the heading of the table. We have added ref32 (now ref 35) to the text as a reference to a recent review of the substrate specificity of DNA glycosylases (line 81) and have corrected the errors in Table 1 indicated by the reviewer and highlighted in yellow in the revised table 1.

Lines 162-163 – The statement with Ref. 107 is not correct. The Ser326Cys OGG1 variant has reduced catalytic activity for 8-oxoG and FapyG compared to the wild-type (see NAR 27, 4001, 1999, and Ref. 32).

  • This statement had been corrected (lines 178-180) and the references suggested by the reviewer have been added.

Lines 273-330 – This section does not describe any of the phenotypes associated with loss of NEIL1 in mice and only anecdotally mentions OGG1 knockout mice – there are significant literatures on these phenotypes, most of which are inside and outside the cancer-specific topics (see e.g., Ref. 162 for several cancer incidences in NEIL1- and NTH1-knockout mice).

  • Here again, the aim of this section was not to provide an extensive list of all the studies in which DNA glycosylases have been knocked out, but instead to give a few examples from each superfamily to illustrate the complexity of such studies and the contrasted effects of knocking out or down DNA glycosylase genes. In particular, we refer to the dual knockout of neil1 and nth1 in mice (ref 162 in the original manuscript, now ref 175) and the increased tumorigenesis associated with these knockouts (see lines 351-353).

Line 351 – The NEIL1 substrate for inhibitor studied was the secondary oxidation products of 8-oxoG. 

  • This has been corrected (lines 420-421).

Line 368-369 – No ref. was given for this statement. Which other DNA glycosylases repair 8-oxoG besides human and other eukaryotic OGG1s? Not NEIL1, NEIL2, NEIL3, NTH1! MutY is a not a part of BER. E. coli Fpg. homolog of OGG1 does remove 8-oxoG.

  • This sentence has been corrected and rephrased (lines 438-440).

In summary, this is a good review of DNA glycosylases as potential targets in cancer therapy. However, it needs an extensive revision as explained in the afore-mentioned comments before it can be considered for publication.

Reviewer 2 Report

Hans et al. present a review providing a detailed overview of the role of DNA glycosyalses (DNA repair enzymes initiating the base excision repair pathway by removing alkylated, degraded and oxidized bases) by focusing on their roles in normal and pathologic contexts, their mode of regulation and their potential as anticancer drug targets.

General comment

This review is interesting and deserves to be published because it complements other recently published reviews on related topics.

Major concern

It is unfortunate that paragraph 5 deals only with the search for DNA glycosylase small inhibitors by high-throughput screening techniques. Why this choice of the authors knowing that many other works have identified inhibitors by low-throughput screening or designed inhibitors on the basis of 3D structural data of DNA glycosylases (included rational computational methods) or from fragment-base design. I think this paragraph should not be reduced to HTS approaches and the works listed below would deserve to be cited and discussed in this paragraph (this list is probably not exhaustive ...). This will bring a real benefit to the review.

- Ung inhibitors

Chung et al. Nat Chem Biol (2009) Impact of linker strain and flexibility in the design of a fragment-based inhibitor

Doi: 10.1038/nchembio.163

Hendricks et al. J. Chem. Information & Modeling (2014) Computational Rationale for the Selective Inhibition of the Herpes Simplex Virus Type 1 Uracil-DNA Glycosylase Enzyme

Doi: 10.1021/ci500375a

Jiang et al. JACS (2005) Uracil-Directed Ligand Tethering: An Efficient Strategy for Uracil DNA Glycosylase (UNG) Inhibitor Development

Doi: 10.1021/ja055846n

Krosky et al. Nucleic Acids Res. (2006) Mimicking damaged DNA with a small molecule inhibitor of human UNG2

Doi:10.1093/nar/gkl747

Wang et al. Nucleic Acids Res. (2016) Using structural-based protein engineering to modulate the differential inhibition effects of SAUGI on human and HSV uracil DNA glycosylase

Doi: 10.1093/nar/gkw185

- NEILs inhibitors

Bliksrud et al. J Inherit Metab Dis (2013) Fumarylacetoacetate inhibits the initial step of the base excision repair pathway: implication for the pathogenesis of tyrosinemia type I

Doi: 10.1007/s10545-012-9556-0

Grin et al. BBRC (2010) Inactivation of NEIL2 DNA glycosylase by pyridoxal phosphate reveals a loop important for substrate binding

Doi: 10.1016/j.bbrc.2010.02.121

Biela et al. Nucleic Acids Res (2014) Zinc finger oxidation of Fpg/Nei DNA glycosylases by 2-thioxanthine: biochemical and X-ray structural characterization

Doi: 10.1093/nar/gku613

Michel et al.  ASC Olega (2019) Computational and Experimental Druggability Assessment of Human DNA Glycosylases

DOI: 10.1021/acsomega.9b00162

Rieux et al. IJMS (2020) Thiopurine Derivative-Induced Fpg/Nei DNA Glycosylase Inhibition: Structural, Dynamic and Functional Insights

Doi: 10.3390/ijms21062058

Minor concern

In several places in the text and table, the authors indicate that NEIL1 excises 8-oxoG. Since the article by Hazra et al. PNAS 2002 (Hazra et al. Identification and characterization of a human DNA glycosylase for repair of modified bases in oxidatively damaged DNA. PNAS (2002) 99(6), 3523–3528; Doi: 0.1073_pnas.062053799) proposing that NEIL1 is able to excise 8-oxoguanine, several authors have not confirmed this result even if this error is repeated in the review by Whitaker et al. (ref 48).

Among the H2TH DNA glycosylases, only the bacterial Fpg is unambiguously a true 8-oxoG-DNA glycosylase. The comparative analysis of the crystal structures of Fpg recognizing 8-oxoG-containing DNA and those of NEIs DNA glycosylases recognizing oxidized pyrimidines, indicates that a large dynamic loop present in Fpg and not in NEIs is a structural motif directly involved in the specific recognition of 8-oxoG (named the 8-oxoG capping loop). Its deletion results in the abolition of the 8-oxoG-DNA glycosylase activity of Fpg but not its capacity to excise certain oxidized pyrimidines as do the NEIs proteins (Duclos et al. Structural and biochemical studies of a plant formamidopyrimidine-DNA glycosylase reveal why eukaryotic Fpg glycosylases do not excise 8-oxoguanine. DNA Repair 11 (2012) 714– 725, Doi: 10.1016/j.dnarep.2012.06.004). In the Hazra article (PNAS 2002), NEIL1 seems capable of excising 8-oxoG when it is placed opposite G or T, mismatches which are not physiological. When 8-oxoG is found opposite C or A (physiological situations before and after replication, respectively), its excision activity for 8-oxoG is anecdotal and not significant from a physiological point of view. Moreover, if NEIL1 were able to excise 8-oxoG opposite A, it would have a mutator effect and this is obviously not the case. 8-oxoG:A is the physiological substrate of MTYH (human MutY homolog), the unique DNA glycosylase specific for the excision of a normal A opposite 8-oxoG.

I therefore urge the authors to remove this error. Instead of ref 48, I would put ref 95.

Author Response

Hans et al. present a review providing a detailed overview of the role of DNA glycosyalses (DNA repair enzymes initiating the base excision repair pathway by removing alkylated, degraded and oxidized bases) by focusing on their roles in normal and pathologic contexts, their mode of regulation and their potential as anticancer drug targets.

General comment

This review is interesting and deserves to be published because it complements other recently published reviews on related topics.

  • We are very grateful for these constructive comments and suggestions that have helped us revise our manuscript.

Major concern

It is unfortunate that paragraph 5 deals only with the search for DNA glycosylase small inhibitors by high-throughput screening techniques. Why this choice of the authors knowing that many other works have identified inhibitors by low-throughput screening or designed inhibitors on the basis of 3D structural data of DNA glycosylases (included rational computational methods) or from fragment-base design. I think this paragraph should not be reduced to HTS approaches and the works listed below would deserve to be cited and discussed in this paragraph (this list is probably not exhaustive ...). This will bring a real benefit to the review.

- Ung inhibitors

Chung et al. Nat Chem Biol (2009) Impact of linker strain and flexibility in the design of a fragment-based inhibitor. Doi: 10.1038/nchembio.163

Hendricks et al. J. Chem. Information & Modeling (2014) Computational Rationale for the Selective Inhibition of the Herpes Simplex Virus Type 1 Uracil-DNA Glycosylase Enzyme. Doi: 10.1021/ci500375a

Jiang et al. JACS (2005) Uracil-Directed Ligand Tethering: An Efficient Strategy for Uracil DNA Glycosylase (UNG) Inhibitor Development. Doi: 10.1021/ja055846n

Krosky et al. Nucleic Acids Res. (2006) Mimicking damaged DNA with a small molecule inhibitor of human UNG2. Doi:10.1093/nar/gkl747

 Wang et al. Nucleic Acids Res. (2016) Using structural-based protein engineering to modulate the differential inhibition effects of SAUGI on human and HSV uracil DNA glycosylase. Doi: 10.1093/nar/gkw185

- NEILs inhibitors

Bliksrud et al. J Inherit Metab Dis (2013) Fumarylacetoacetate inhibits the initial step of the base excision repair pathway: implication for the pathogenesis of tyrosinemia type I. Doi: 10.1007/s10545-012-9556-0

Grin et al. BBRC (2010) Inactivation of NEIL2 DNA glycosylase by pyridoxal phosphate reveals a loop important for substrate binding. Doi: 10.1016/j.bbrc.2010.02.121

Biela et al. Nucleic Acids Res (2014) Zinc finger oxidation of Fpg/Nei DNA glycosylases by 2-thioxanthine: biochemical and X-ray structural characterization. Doi: 10.1093/nar/gku613 

Michel et al.  ASC Olega (2019) Computational and Experimental Druggability Assessment of Human DNA Glycosylases. DOI: 10.1021/acsomega.9b00162

Rieux et al. IJMS (2020) Thiopurine Derivative-Induced Fpg/Nei DNA Glycosylase Inhibition: Structural, Dynamic and Functional Insights. Doi: 10.3390/ijms21062058

  • We have now included a description of these alternative approaches for the identification of DNA glycosylase inhibitors as part of section 5 (lines 374-411).

Minor concern

In several places in the text and table, the authors indicate that NEIL1 excises 8-oxoG. Since the article by Hazra et al. PNAS 2002 (Hazra et al. Identification and characterization of a human DNA glycosylase for repair of modified bases in oxidatively damaged DNA. PNAS (2002) 99(6), 3523–3528; Doi: 0.1073_pnas.062053799) proposing that NEIL1 is able to excise 8-oxoguanine, several authors have not confirmed this result even if this error is repeated in the review by Whitaker et al. (ref 48).

Among the H2TH DNA glycosylases, only the bacterial Fpg is unambiguously a true 8-oxoG-DNA glycosylase. The comparative analysis of the crystal structures of Fpg recognizing 8-oxoG-containing DNA and those of NEIs DNA glycosylases recognizing oxidized pyrimidines, indicates that a large dynamic loop present in Fpg and not in NEIs is a structural motif directly involved in the specific recognition of 8-oxoG (named the 8-oxoG capping loop). Its deletion results in the abolition of the 8-oxoG-DNA glycosylase activity of Fpg but not its capacity to excise certain oxidized pyrimidines as do the NEIs proteins (Duclos et al. Structural and biochemical studies of a plant formamidopyrimidine-DNA glycosylase reveal why eukaryotic Fpg glycosylases do not excise 8-oxoguanine. DNA Repair 11 (2012) 714– 725, Doi: 10.1016/j.dnarep.2012.06.004). In the Hazra article (PNAS 2002), NEIL1 seems capable of excising 8-oxoG when it is placed opposite G or T, mismatches which are not physiological. When 8-oxoG is found opposite C or A (physiological situations before and after replication, respectively), its excision activity for 8-oxoG is anecdotal and not significant from a physiological point of view. Moreover, if NEIL1 were able to excise 8-oxoG opposite A, it would have a mutator effect and this is obviously not the case. 8-oxoG:A is the physiological substrate of MTYH (human MutY homolog), the unique DNA glycosylase specific for the excision of a normal A opposite 8-oxoG.

I therefore urge the authors to remove this error. Instead of ref 48, I would put ref 95.

  • This has been corrected in the revised manuscript (see Table 1) and the proposed reference has been added.

Reviewer 3 Report

In their review, Hans et al. discuss DNA glycosylases and the prospect of targeting these proteins for cancer therapy. I find this area a very interesting topic. However the authors may be presenting too broad of a topic. This might be preventing them from providing a compelling summary of anything specific. I would consider narrowing the focus to only one family of enzymes, or type of DNA damage. My comments focus on uracil BER which is my expertise – I do not find that the potential of uracil BER as an anti-cancer target is presented fairly because there is no depth to the material presented.

Page 2 Line 67: I do not believe that there are only 3 ways in which base lesions are generated in cancers. How do you define a lesion? Uracil is often considered a lesion. Accidental misincorporation of uracil by a polymerase does not fit any of the 3 ways presented.

Figure 1 and general comment: There is more than one BER pathway. I believe that the authors are presenting what is commonly called ‘short patch BER’ but do not discuss ‘long patch BER’.

Page 4 Line 122: TDG is indeed recognized as an epigenetic mediator. However I am not sure I agree that is commonly considered its main function.

Table 1: Some UNG2 PTMs are not listed (acetylation: Iveland, 2020 J Transl Med; Bao 2020 Free Rad Biol Med).

Table 1: Simply listing reported PTMs without discussing their importance is not helpful

Page 6 line 191: UNG2 and TDG are not ‘apparently redundant’. TDG is dsDNA specific for U/G (Not U/A). UNG2 can remove U in any context, dsDNA and ssDNA

Figure 2: I am not sure that the significance of this Figure was really elucidated in the manuscript

Page 8 line 215: “Generally phosphorylation has been shown to increase the catalytic…” This sentence is an oversimplification and in many cases not true. The authors are also simplifying how modifications can either affect the intrinsic function of an enzyme, or the modifications can affect the enzyme interaction with other proteins that affect its activity.

As an example, For UNG2, there are examples of phosphorylation having no effect on the enzyme intrinsic activity (Weiser, 2017, Biophys j). But this modification might affect UNG2 interaction with certain binding partners. Alternatively these modifications affect UNG2 stability inside the cell

Page 8 line 234: UNG2 is not regulated by a ‘balance’ of phosphorylation and ubiquitination. Rather, successive phosphorylations lead to ubiquitination through the formation of a degron (Hagen, 2008)

Page 8 line 240: This article where SUMO is overexpressed is not very clear how SUMO is affecting UNG2 expression- I am not sure if it is clear whether their fluorescent SUMO protein is acting as  a PTM

Page 8 line 248: XRCC1 is a very poorly characterized binding partner of UNG2 and is not biophysically validated. Its common binding partners PCNA and RPA are not discussed.

Page 9 line 297: The role of UNG2 in thymidylate synthase inhibitor toxicity is very complicated and should include 5-Fluorouracil as well as pemetrexed. This topic cannot be covered in one paragraph

The entire realm of BER and uracil bases in somatic hypermutation/class switch recombination is not addressed- this is critical as their role in B-cell lymphomas has been clearly demonstrated

Author Response

In their review, Hans et al. discuss DNA glycosylases and the prospect of targeting these proteins for cancer therapy. I find this area a very interesting topic.

  • We thank the reviewer for these constructive comments and suggestions that have helped us revise our manuscript.

However the authors may be presenting too broad of a topic. This might be preventing them from providing a compelling summary of anything specific. I would consider narrowing the focus to only one family of enzymes, or type of DNA damage.

  • We understand the reviewer’s point of view, but feel that focusing on a single family of DNA glycosylases or on a single type of DNA damage would not provide the desired overview of DNA glycosylases as potential drug targets in cancer. With this review, we have chosen to illustrate the importance of these enzymes as anticancer drug targets by presenting different examples from each of the different glycosylase superfamilies, and not an exhaustive account, of the complex involvement and regulation of DNA glycosylases in cancer cells.

My comments focus on uracil BER which is my expertise – I do not find that the potential of uracil BER as an anti-cancer target is presented fairly because there is no depth to the material presented.

Page 2 Line 67: I do not believe that there are only 3 ways in which base lesions are generated in cancers. How do you define a lesion? Uracil is often considered a lesion. Accidental misincorporation of uracil by a polymerase does not fit any of the 3 ways presented.

  • This sentence has been rephrased (lines 67-71).

Figure 1 and general comment: There is more than one BER pathway. I believe that the authors are presenting what is commonly called ‘short patch BER’ but do not discuss ‘long patch BER’.

  • A more thorough description of BER has now been included in the revised manuscript, which distinguishes short-patch and long-patch repair (lines 92-102). Figure 1 has also been edited to illustrate these two sub-pathways.

Page 4 Line 122: TDG is indeed recognized as an epigenetic mediator. However I am not sure I agree that is commonly considered its main function.

  • This sentence has been modified (lines 136-138).

Table 1: Some UNG2 PTMs are not listed (acetylation: Iveland, 2020 J Transl Med; Bao 2020 Free Rad Biol Med).

  • The table has been updated with the missing elements and associated references.

Table 1: Simply listing reported PTMs without discussing their importance is not helpful

  • We do not simply list the PTMs – we give a list in Table 1, but in the main text we give detailed examples of the different types of PTMs found in DNA glycosylases and their role in regulating DNA glycosylase function in order to highlight their potential as novel drug targets (see lines 242-286).

Page 6 line 191: UNG2 and TDG are not ‘apparently redundant’. TDG is dsDNA specific for U/G (Not U/A). UNG2 can remove U in any context, dsDNA and ssDNA

  • This sentence has been modified (lines 210-211).

Figure 2: I am not sure that the significance of this Figure was really elucidated in the manuscript

  • In the revised manuscript, we have now highlighted the most pronounced changes in DNA glycosylase expression levels observed in cancer tissues, which are shown in Figure 2, and relate these mRNA expression levels to carcinogenesis and in some cases to patient outcome (lines 212-226).

Page 8 line 215: “Generally phosphorylation has been shown to increase the catalytic…” This sentence is an oversimplification and in many cases not true. The authors are also simplifying how modifications can either affect the intrinsic function of an enzyme, or the modifications can affect the enzyme interaction with other proteins that affect its activity.

As an example, For UNG2, there are examples of phosphorylation having no effect on the enzyme intrinsic activity (Weiser, 2017, Biophys j). But this modification might affect UNG2 interaction with certain binding partners. Alternatively these modifications affect UNG2 stability inside the cell.

  • This has been corrected and a more thorough description of the effects of phosphorylation on the activity of UNG2 and its interaction with protein partners has now been provided (lines 247-258).

Page 8 line 234: UNG2 is not regulated by a ‘balance’ of phosphorylation and ubiquitination. Rather, successive phosphorylations lead to ubiquitination through the formation of a degron (Hagen, 2008)

  • The sentence has been modified accordingly (lines 274-277).

Page 8 line 240: This article where SUMO is overexpressed is not very clear how SUMO is affecting UNG2 expression- I am not sure if it is clear whether their fluorescent SUMO protein is acting as  a PTM

  • This sentence has been modified to provide a more accurate account of this study in HepG2 cells (lines 284-286).

Page 8 line 248: XRCC1 is a very poorly characterized binding partner of UNG2 and is not biophysically validated. Its common binding partners PCNA and RPA are not discussed.

  • This sentence has been corrected (line 290 and 292).

Page 9 line 297: The role of UNG2 in thymidylate synthase inhibitor toxicity is very complicated and should include 5-Fluorouracil as well as pemetrexed. This topic cannot be covered in one paragraph

  • We agree with the reviewer that it is difficult to summarize this topic in one paragraph and have thus decided to remove this example to avoid any oversimplification, which would be counterproductive.

The entire realm of BER and uracil bases in somatic hypermutation/class switch recombination is not addressed- this is critical as their role in B-cell lymphomas has been clearly demonstrated

  • We have added a couple of sentences describing the role of UNGs in somatic hypermutation and class switch recombination (lines 133-136). As for the role of UNG2 in thymidylate synthase inhibitor toxicity, we feel that an in-depth description of the role of UNG in AID-overexpressing B-cell lymphomas is beyond the realm of this review.

Round 2

Reviewer 1 Report

This revised manuscript can now be accepted for publication. However, many references including refs. 37, 38, 73, 107, 117, 125, 135, 138, 160 and others are missing page numbers (all pages or last page). All references should be carefully checked and corrected where necessary.